# Anisotropic ESCRT-III architecture governs helical membrane tube formation

Joachim Moser von Filseck [1], Luca Barberi [2,9✉], Nathaniel Talledge [3,4,5,10], Isabel E. Johnson [3,4], Adam Frost [3,4,5,6], Martin Lenz [2,7] & Aurélien Roux [1,8✉]

ESCRT-III proteins assemble into ubiquitous membrane-remodeling polymers during many cellular processes. Here we describe the structure of helical membrane tubes that are scaffolded by bundled ESCRT-III filaments. Cryo-ET reveals how the shape of the helical membrane tube arises from the assembly of two distinct bundles of helical filaments that have the same helical path but bind the membrane with different interfaces. Higher-resolution cryo-EM of filaments bound to helical bicelles confirms that ESCRT-III filaments can interact with the membrane through a previously undescribed interface. Mathematical modeling demonstrates that the interface described above is key to the mechanical stability of helical membrane tubes and helps infer the rigidity of the described protein filaments. Altogether, our results suggest that the interactions between ESCRT-III filaments and the membrane could proceed through multiple interfaces, to provide assembly on membranes with various shapes, or adapt the orientation of the filaments towards the membrane during membrane remodeling.

[1] Biochemistry Department, University of Geneva, 1211 Geneva, Switzerland. [2] LPTMS, CNRS, Université Paris-Sud, Université Paris-Saclay, 91405 Orsay, France. [3] Department of Biochemistry and Biophysics, University of California, San Francisco, San Francisco, CA 94158, USA. [4] California Institute for Quantitative Biosciences, San Francisco, CA 94158, USA. [5] Department of Biochemistry, University of Utah, Salt Lake City, UT 841112, USA. [6] Chan Zuckerberg Biohub, San Francisco, CA 94158, USA. [7] Laboratoire de Physique et Mécanique des Milieux Hétérogènes, UMR 7636, CNRS, ESPCI Paris, PSL Research University, Université Paris Diderot, Sorbonne Université, 75005 Paris, France. [8] Swiss National Centre for Competence in Research Programme Chemical Biology, Geneva 1211, Switzerland. [9] Present address: Biochemistry Department, University of Geneva, 1211 Geneva, Switzerland. [10] Present address: Institute for Molecular Virology, University of Minnesota-Twin Cities, Minneapolis, MN 55455, USA. ✉email: luca.barberi@unige.ch; aurelien.roux@unige.ch

The Endosomal Sorting Complexes Required for Transport (ESCRT)-III proteins are an evolutionarily ancient family of proteins that execute membrane scission in different cellular contexts (reviewed in ref. [1]). ESCRT-III can polymerize into rings and spirals in solution[2–4] or on membrane substrates[5,6]. When single or several ESCRT-III proteins are incubated with model membranes in vitro or over-expressed in cells, they deform membranes into straight and conical tubes[6,7], demonstrated in most detail by the formation of tubules by CHMP1B alone and in complex with IST1/CHMP8[6]. Similar but inverted conical structures are also observed in vivo by overexpression of CHMP4A/B[7] and at the neck of budding Gag envelopes[8]. Dynamics of ESCRT-III assembly also suggest that single assemblies in MVB biogenesis in yeast are compatible with spirals or cones[9]. Mechanistically, we have previously shown that flat spirals formed on lipid membranes from the ESCRT-III protein Snf7 can accumulate elastic energy, and that this energy could be channeled to shape a flat membrane into a tube through a buckling transition[10]. However, Snf7 spirals fail to deform artificial membranes in vitro[5]. This could be due to the high flexibility of Snf7 polymers[2,5], which do not provide enough force to deform the membrane. In this case, rigidification of the filament through the binding of additional subunits could trigger buckling. Importantly, Snf7 forms flat spirals on membranes[3] and in solution[2], which indicates that Snf7 filaments only present spontaneous curvature and no torsion. In such circumstances, binding of additional subunits, which induce a twist in the co-filaments and lead to the formation of a helical structure, could induce buckling. Indeed, recruitment of Vps24/Vps2 to flat Snf7 spirals, via electrostatic interaction between Snf7 helix α4 and Vps24[11], leads to the formation of helical structures without membrane inside[3]. We have previously shown that the addition of Vps24/Vps2 to a membrane-bound Snf7 filament leads to the formation of a second, parallel strand next to the Snf7 filament[12], and the formation of such a composite polymer may trigger buckling. In this report, we show that the addition of Vps24/Vps2 to membrane-bound Snf7 in vitro does indeed induce a membrane shape transition from flat to tubes. Surprisingly, however, this transition does not result in straight, cylindrical tubes scaffolded by a helical polymer, but in membrane tubes that are shaped like hollow corkscrews, hereafter referred to as helical tubes. We show using cryogenic electron tomography (cryo-ET) and subtomogram averaging (STA) that this unusual structure is supported by an unexpected protein-membrane binding scheme, involving two different membrane-binding interfaces. We further demonstrate through physical modeling that the stability of this architecture implies that the two corresponding binding energies are significantly different. In addition, we obtain a higher-resolution structure of the helical polymers, bound to a helical bicelle ribbon, confirming that the Snf7/Vps24/Vps2 copolymer has a twist and binds the membrane in orientations different from those previously published. The dimensions of these non-constrained helical polymers differ slightly from those observed on the helical tubes. These differences suggest that helical protein polymers are under elastic stress in the helical tubes. Finally, by comparing the morphology of the helical tubes to those of the bicelle-bound copolymers, we infer the binding energy difference, as well as the stiffness of the ESCRT-III copolymers. Our results are consistent with the notion that dynamic changes in polymer-membrane interactions coupled with high bending and torsional rigidities in the copolymer are essential to trigger a buckling transition.

## Results

### Helical tubulation of liposomes by ESCRT-III heteropolymers.
To test whether binding of Vps24 and Vps2 to Snf7 spirals could induce a membrane shape transition, we incubated liposomes with recombinant Snf7 until they were decorated by flat Snf7 spirals[5], then added recombinant Vps24 and Vps2 and incubated the mixture for several hours. Using negative stain electron microscopy (EM), we observed vesicles decorated with flat spirals (Fig. 1a)[5,12] and helical tubes that were decorated with filamentous protein polymers (Fig. 1b, c). Cryogenic electron microscopy (cryo-EM) of helical tubes confirmed that they consisted of protein filaments bound to an open helical membrane tube (Fig. 1d–f), and that their regularity made them amenable to higher-resolution imaging. Further investigation of these helical tubes revealed that they only form in the presence of all three proteins (Supplementary Fig. 1a–c). They had an average diameter of 23.9 ± 3.7 nm and were coiled into a helix with an outer diameter of 82.3 ± 6.1 nm and a pitch of 53.1 ± 7.6 nm (all values average ± SD; Supplementary Table 1) (Supplementary Fig. 1d–g). Their prevalence increased with protein concentrations and incubation time, indicating thermodynamic stability. Helical tubes are an unusual membrane shape, as their high curvature makes them a priori energetically unfavorable compared to other shapes, yet assemblies of different human ESCRT-III proteins on liposomes can generate similar deformations[13]. To understand the origin of their stability, we aimed to characterize their structural determinants in more detail.

To visualize the ESCRT-III filament organization around the helical tubes, we performed cryo-ET on vitrified helical membrane tubes and used image filtering and manual segmentation on reconstructed tomographic volumes. All tubes appeared as left-handed helices, although we cannot confirm that this is the correct handedness without a chiral internal standard. On the surface of the tubes, we observed six to eight filaments parallel to the tube axis forming multi-stranded bundles (Fig. 1g–i, Supplementary Fig. 1h–j, Movies 1–2). The filaments were almost always excluded from the inside of the tube helix and had the same thickness as negatively stained, double-stranded Snf7/Vps24/Vps2 heteropolymers (4.9 ± 0.5 nm; average ± SD)[12]. From this, we concluded that the peculiar organization of the filaments around the tube must minimize the energy of the helical membrane shape.

Helical membrane cylinders have been reported before: cylindrical stacks of lipid membranes remodel into helical tubes in the presence of specific membrane-binding polymers, and it was suggested that the shape could emerge from gradients of spontaneous curvature across the membrane[14]. Helical membrane tubes have also been predicted in the presence of curved polymers whose membrane-binding interface is not located within the polymer's groove (like in BAR domains), but on the orthogonal side[15]. We hypothesize that a similar mechanism determines the emergence of helical tubes in our experiments. Indeed, if Snf7/Vps24/Vps2 helical filaments preferred binding the membrane along their spontaneous direction of curvature, we would expect them to shape straight membrane tubes, as it happens with BAR domain-containing proteins or dynamin-coated membrane tubes[16]. Since we did not observe straight tubes in our experiments, we hypothesized that Snf7/Vps24/Vps2 helical filaments force the tube to follow an equilibrium helical path because they prefer to bind the membrane perpendicular to their spontaneous direction of curvature. We further develop this argument through mathematical modeling of the helical tubes.

**Two distinct ESCRT-III filament bundles on helical tubes**. To obtain a more detailed understanding of how the filaments are organized, we performed STA on slices along the membrane tube axis. The variability in tube dimensions in the dataset made it impossible to resolve the entire tube. We, therefore, focused on the filaments on the outer tube surface and obtained a ~32 Å reconstruction (Fig. 2). This map revealed that the filaments

LUVs + Snf7 + Vps2 + Vps24

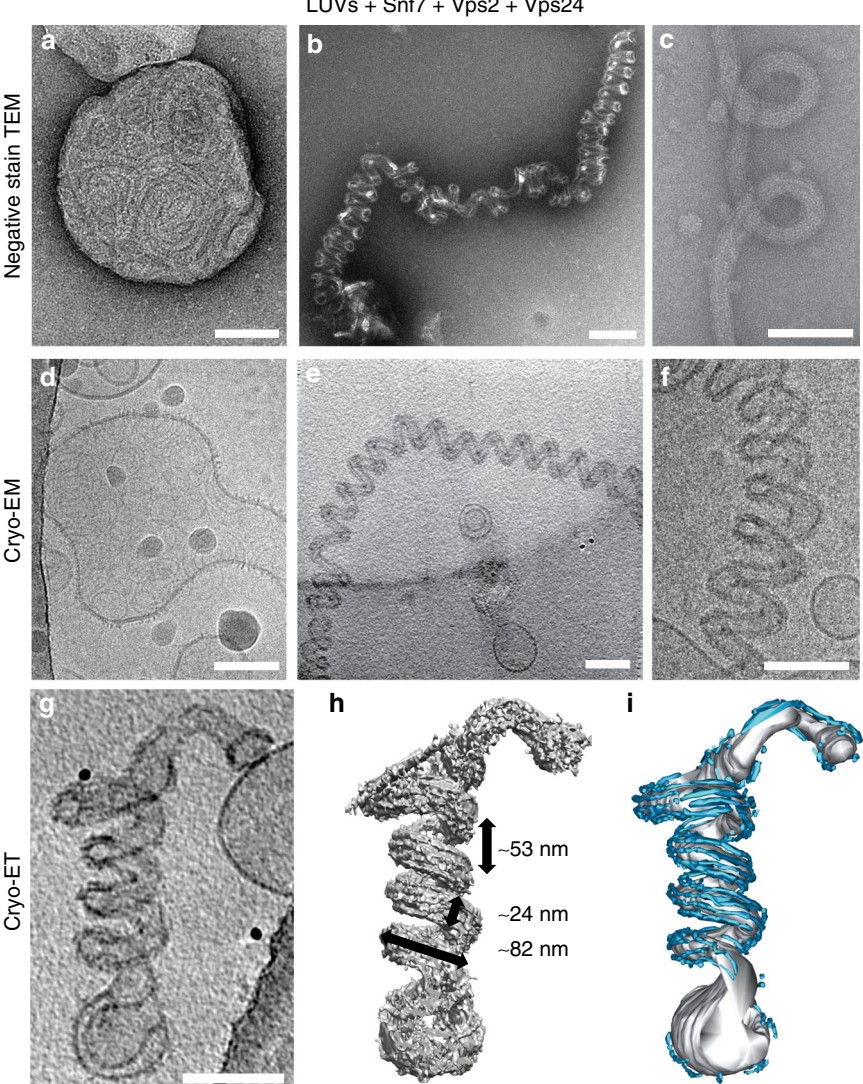

**Fig. 1 Helical tubulation of liposomes by ESCRT-III heteropolymers.** Electron micrographs showing undeformed liposomes (**a**, **d**) and helical membrane tubes (**b**, **c**, **e**, **f**) decorated with Snf7/Vps24/Vps2 on negatively stained (**a**–**c**) and vitrified (**d**–**f**) samples. The reconstructed cryo-ET volume of a helical membrane tube projected in Z (**g**) and volume view after filtering (**h**) or manual segmentation (**i**) showing the organization of protein filaments (cyan) along the helical membrane tube axis (gray). The helical tube dimensions are rounded averages, for details see Supplementary Table 1. All scale bars 100 nm. Source data are provided as a Source Data file.

cluster in three separate regions with two clearly defined grooves between them (Fig. 2a, Supplementary Fig. 2a). The central cluster, containing two filaments, covered a 13 nm wide region around the equator of the tube (equatorial filaments, blue). Two additional filament clusters, each containing 2–3 filaments, were shifted up and down from the equator, respectively, (polar filaments, red) and appeared wider (16–20 nm) (Fig. 2b–d). The resolution of the shifted, polar filaments was limited as their positions varied more with tube diameter compared to equatorial filaments (Supplementary Table 1).

With further STA focused on the equatorial cluster, we reconstructed a focused map of this area (~32 Å resolution), revealing that the two equatorial filaments contained two strands each (Fig. 2e–g, Supplementary Fig. 2b). The filaments bundled in a plane parallel to the tube's helical axis and their membrane binding area was on the bundle's inside, also parallel to the helical axis, as observed in previously described ESCRT-III heteropolymers[6]. Yet, in our case, both strands appeared to be interacting with the membrane. The filaments in the polar

clusters, based on their width, could be double-stranded as well, though our reconstructions were unable to resolve the sub-structure directly. In contrast to the equatorial filaments, however, the bundling plane of the polar filament strands was perpendicular to the helical axis, as was its membrane-binding interface (Fig. 2h). This orientation fits the double-stranded spirals formed by Snf7/Vps24/Vps2 on flat bilayers[12]. Overall, the architectures of equatorial and polar filaments appeared to be similar: both were composed of at least two double-stranded filaments, bundled together as a helical ribbon along the surface of the tube. However, the geometry of the helical tube makes it impossible that all filaments have the same path and bind the membrane with the same interface (Fig. 2h). For the same reasons, interactions between filaments within a bundle cannot be the same within polar filaments and equatorial filaments. Given that helical tubes did not form in the absence of any of the three ESCRT-III subunits (Snf7, Vps24, and Vps2), we conclude that both kinds of filaments were formed from all three proteins. At this resolution, however, we cannot determine whether polar and

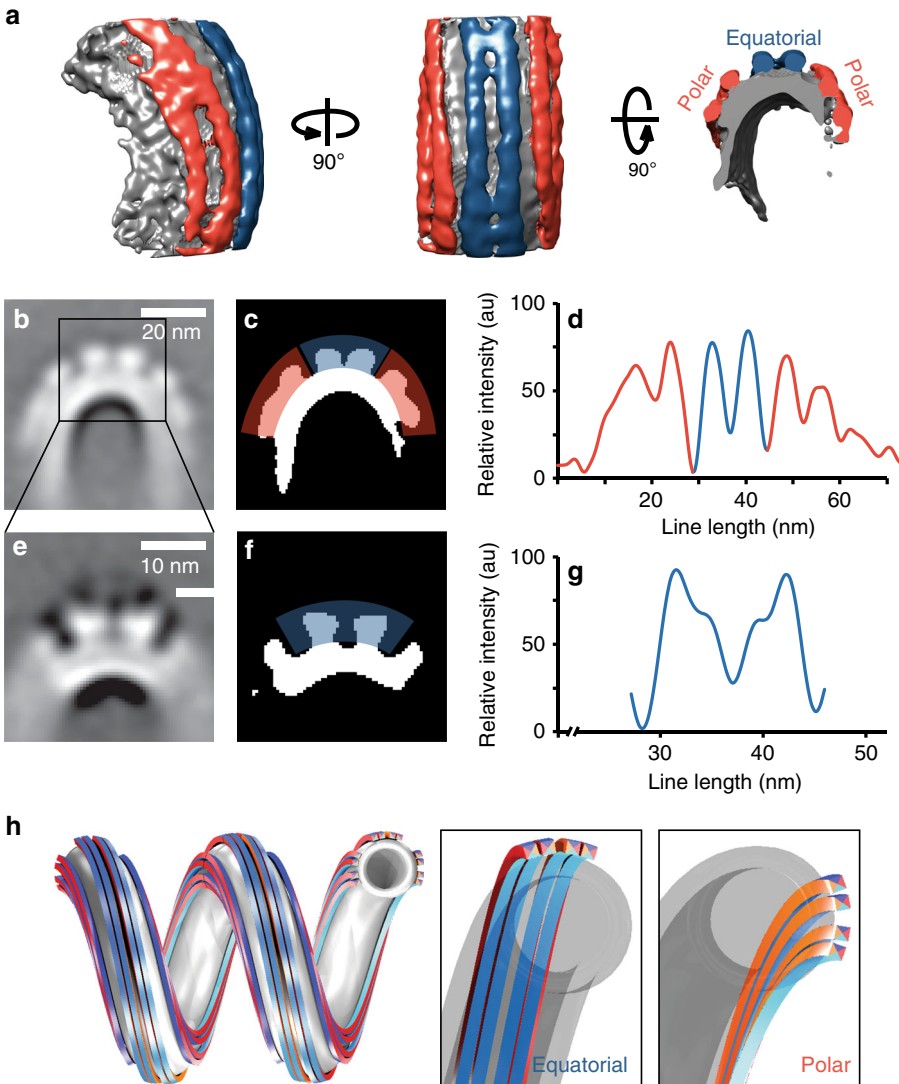

**Fig. 2 ESCRT-III filament bundles form distinct clusters on the surface of helical tubes. a** Side view (left), top view (center) and cross-section (right) of a global subtomogram average showing filaments following the tube axis, in the equatorial (blue) and polar (red) binding mode, respectively. **b** Sum projection of a central segment of the tube in **a** showing filaments on the outer surface of the helical tube, organized as one equatorial and two polar clusters. Scale bar 20 nm. **c** Equatorial (blue) and polar (red) filament cluster highlighted on the thresholded image **b**. **d** Intensity profile of protein density in **c**. **e** Projection of the refined map of the equatorial cluster showing that both filaments of the cluster are made of two strands each. Scale bar 10 nm. **f** Thresholded image of **e**. **g** Intensity profile of protein density in **e**. For filament dimensions, see Supplementary Table 1. **h** 3D model of one equatorial and two polar filament bundles, each formed from two double-strands, on a helical membrane tube (gray). All filaments are identical, except that equatorial and polar filaments bind the membrane through the cyan and orange interfaces, respectively (insets). Filaments in the two hemispheres are shown as antiparallel. Source data are provided as a Source Data file.

equatorial filaments contain different subunit compositions or stoichiometries. Different examples of ESCRT-III copolymers made with different subunits, like CHMP1B and IST1/CHMP8, have very different spontaneous curvatures and shapes[6,17]. Our equatorial and polar filaments did have similar helical paths, bundling properties and dimensions, though, leading us to favor the hypothesis that the polar and equatorial filaments comprised the same subunits at similar stoichiometry. While the possibility that ESCRT-III molecules bind their target membranes with two different orientations seems a priori unexpected, existing structural studies have reported different membrane binding interfaces for Snf7 versus CHMP1B[6,17,18].

**Organization of tube-less ESCRT-III filaments**. To clarify the interplay between the elasticity of the ESCRT-III filaments and

that of the membrane in determining the shape of the helical tube, we sought to analyze the spontaneous shape of ESCRT-III filaments without a helical membrane tube for higher-resolution imaging. When incubating Snf7/Vps24/Vps2 with detergent-solubilized lipids, different helical ribbons formed without complete membrane tubes during detergent removal by dilution (Supplementary Fig. 3a–c). We suppose that the detergent removal generates a great number of small membrane structures that nucleate ESCRT-III filaments that self-assemble along a bicelle ribbon. Most of these tube-less, helical ribbons assembled into sharp zigzag shapes (Fig. 3a, red arrows in Supplementary Fig. 3a–c), a smaller population appeared sinusoidal (Fig. 3b, blue arrows in Supplementary Fig. 3a–c), and a third population displayed significantly larger ribbons with varying strand numbers and diameters (Fig. 3c, yellow arrows in Supplementary

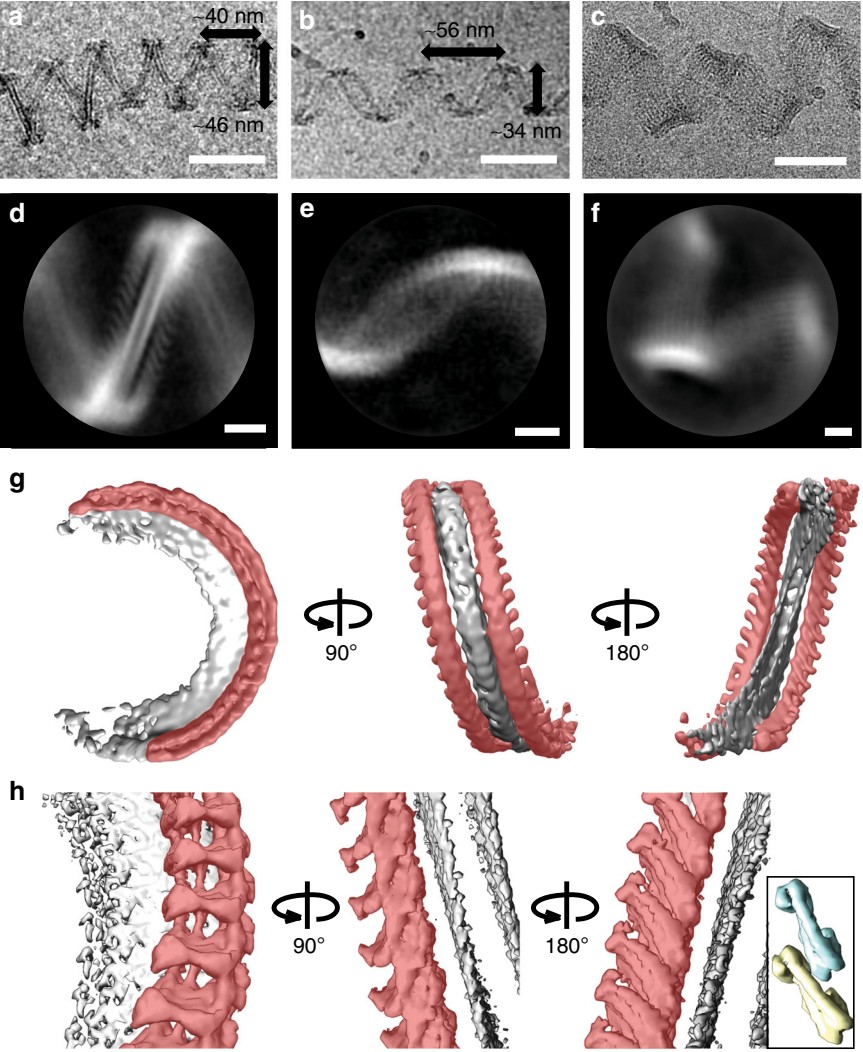

**Fig. 3 Organization of tube-less ESCRT-III filaments.** Electron micrographs (**a–c**, scale bars 50 nm) and 2D class averages (**d–f**, scale bars 10 nm) showing different tube-less, helical ESCRT-III filament bundles formed from Snf7, Vps24, and Vps2 upon detergent removal. The majority of ribbons adopted a zigzag shape (**a**, **d**), others appeared sinusoidal (**b**, **e**) and a third set consisted of helical ribbons with higher strand numbers (**c**, **f**). The ribbon dimensions are rounded averages, for details see Supplementary Table 1. **g** Unmasked 3D average of **a**, **d** shows that the center of the ribbon is a helical bicelle with its plane perpendicular to the tube axis (grey). There are two anti-parallel double-stranded filaments on both sides of the bicelle (red). **h** 3D average as in **a**, **c** with an asymmetric mask that included only one double-stranded filament (red). Inset: scale-matched densities corresponding to two closed-conformation IST1/CHMP8 subunits from EMD-6461[6] are shown for comparison. Source data are provided as a Source Data file.

Fig. 3a–c). We did not observe any of these assemblies if any of the three ESCRT-III subunits was omitted (Supplementary Fig. 3d–f). We used single-particle 2D and 3D averaging approaches to analyze these tube-less helical protein filament ribbons and determined 2D class averages (Fig. 3d–f). The overall appearance of the sinusoidal ribbons suggested that they comprise multi-stranded filaments oriented along a helical path similar to that of the equatorial filaments we observed bound to the helical membrane tubes (pitch $55.7 \pm 8.5$ nm; diameter = $34.1 \pm 5.0$ nm, width $13.6 \pm 2.1$ nm average ± SD; Supplementary Table 1) (Fig. 3b–e).

Analysis of the more ordered zigzag structure (Fig. 3a–d) led to a 3D reconstruction at ~15 Å resolution. This structure revealed a helical ramp formed around a bicelle, a tension-less lipid bilayer stabilized by detergents, with the bicelle plane oriented perpendicular to the helix axis. Given that such helical bicelles cannot form from vesicles, this explains why we only observed ESCRT-III ribbons with initially detergent-solubilized lipids. On both sides of the bicelle, we observed filamentous polymers with subunit

dimensions consistent with other double-stranded ESCRT-III structures[6,11,12]. The observed pitch ($39.8 \pm 6.9$ nm; diameter = $46.2 \pm 4.9$ nm; average ± SD; Supplementary Table 1) of the filament indicated a significantly elevated torsion and/or torsional rigidity compared to other helical ESCRT-III polymers[4,6]. Considering the apparent subunit tilt on both sides of the bicelle, the filaments appeared to be anti-parallel to each other (Fig. 3d–g).

We confirmed the anti-parallel orientation of the two polymers by a 3D reconstruction at a higher resolution (~11 Å) that was computed by using masks to focus on one side of the bicelle only (Fig. 3h). The subunits appeared to polymerize in the same way as previously described ESCRT-III heteropolymers[6], and were oriented along a similar helical path. Surprisingly, both strands seemed to interact with the membrane, and their membrane-binding interface was oriented perpendicular to the main helical axis (Fig. 3g, h). The interface was therefore perpendicular to that postulated for CHMP1B, which was parallel to the helix axis[6]. Molecular docking allowed fitting both filaments with crystal structures of subunits in the open (*D. melanogaster* CHMP4B

homolog Shrub, PDB 5J45[19]; yeast Snf7, PDB 5FD9[18]) and closed conformation (Human CHMP3; PDB 3FRT[20]), respectively (Supplementary Fig. 3d), with inter-subunit connectivity consistent with known ESCRT-III heteropolymer structures[6]. The resolution of the map, however, did not allow us to discern the identities or unambiguous conformations for the subunits of either strand. Nevertheless, the zigzag tube-less ribbon's dimensions and architecture are compatible with the polar filaments on helical tubes, and confirmed that the polar filaments of the helical tube are also double-stranded. These results demonstrate that ESCRT-III filaments can bind the membrane with a previously undescribed orientation perpendicular to that of their curvature.

**Mathematical model of helical tubes' mechanical equilibrium.** To understand the roles of ESCRT-III filament properties in shaping the membrane into helical tubes, we developed mathematical models that describe the competition between filament and membrane rigidities, membrane tension and filament-membrane binding energy. Here we summarize our conclusions, and refer the reader to the Supplementary Information for detailed derivations.

In a first approach, we show that the membrane-binding interface observed in polar filaments (Fig. 3) is not only compatible with the existence of helical membrane tubes, but is actually required for their stability. To understand this requirement, we consider that the helical tube is not the only membrane structure compatible with the helical structure of Snf7/Vps24/Vps2 helical filaments: such filaments could, hypothetically, also enclose a straight membrane tube, implying a much smaller membrane bending energy cost. However, this alternative structure would imply that all filaments bind in their equatorial mode, as opposed to the mixed equatorial and polar binding observed on helical tubes. We thus interpret the formation of helical tubes as opposed to straight tubes as evidence that the polar filaments' binding mode is energetically more favorable than that of the equatorial filaments, and that it more than compensates for the higher membrane curvature energy of helical tubes. To turn this reasoning into a quantitative estimate of the minimal binding energy difference between polar and equatorial filaments, we developed a mathematical model to compute the deformation energy of a flexible membrane of tension $\sigma$ and stiffness $\kappa$ enclosed by a non-deformable helical scaffold of radius $R$ and pitch $2\pi P$. This choice of a fixed radius and pitch is consistent with the modest filament deformation induced by the presence of membrane tubes, compared to their tube-less shape.

We compare the energies of a helical tube and a straight tube under the assumption that two filament binding modes differ by an energy $\mu$ per unit filament length, where $\mu > 0$ promotes polar filaments over equatorial ones and thus favors helical tubes. The relative stability of either configuration depends on two dimensionless parameters, which we use as coordinates for the phase diagram (Fig. 4a): the rescaled membrane tension $\sigma R^2/\kappa$ and the rescaled differential binding energy per filament length $\mu R/\kappa$. We find that helical tubes are always favored at high rescaled membrane tension, and that lowering $\sigma R^2/\kappa$ leads to an increase of the membrane tube radius $r$, with different outcomes depending on the value of the rescaled differential binding energy per filament length. For high values of $\mu R/\kappa$, helical tubes remain stable at all $\sigma R^2/\kappa$. For lower values of $\mu R/\kappa$, $r$ increases significantly before reaching a $\mu R/\kappa$-dependent critical value $r_c$ where the system transitions to a straight tube (Fig. 4b). While the surface tension $\sigma$ of the membrane is not directly experimentally accessible in our setup, this reasoning demonstrates that given a certain value of $\mu R/\kappa$, helical tubes with radii larger than $r_c(\mu R/\kappa)$ cannot occur. Consequently, our observation of relatively thick tubes with average radius $r_{exp} = 12.1$ nm implies that $r_c\left(\frac{\mu R}{\kappa}\right) > r_{exp}$, which, according to our calculations, implies that the membrane-binding energy difference between polar and equatorial Snf7/Vps24/Vps2 filaments is larger than or equal to $2k_BT$ per monomer. This value is compatible with the previously estimated membrane-binding energy per monomer of Snf7 polymers alone (about 4 $k_BT$)[5], suggesting that Vps24 and Vps2 may be significant contributors of the binding of ESCRT-III filaments to lipid membranes.

In a second approach, we look more closely at the deformation of the helical filaments. We thus relax the assumption of a non-deformable helical scaffold and endow the Snf7/Vps24/Vps2 filaments with bending and torsional rigidities, characterized by the filaments' bending and torsional persistence lengths, $\ell_p$ and $\ell_t$, respectively. We furthermore define the helical parameters (radius and pitch) of zigzag-shaped (Fig. 3a–d) and sinusoidal (Fig. 3b–e) tube-less filaments, respectively, as resting conformations of polar and equatorial filaments (Fig. 2a), respectively, on helical tubes (Supplementary Table 1). As a result of their deformability, enclosing a helical membrane tube inside our model filaments results in a variation of their radius and pitch. By matching these predicted variations to the observed differences in filament radius and pitch between the tube-less situation (Fig. 3) and the tube-enclosing configurations of (Figs. 1, 2) as the result of the membrane, we establish a lower bound $\ell_p \geq \ell_p^{\min} = 114$ nm

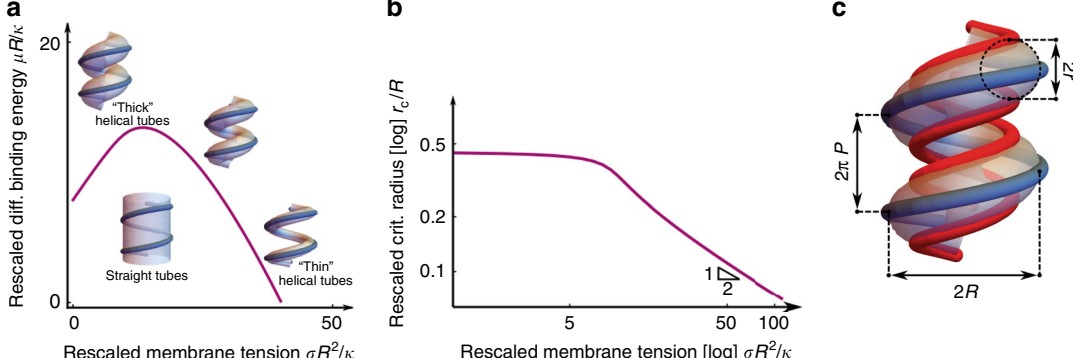

**Fig. 4 Mathematical modeling of helical tubes' mechanical equilibrium. a** Phase diagram showing the energetically favored shape between straight and helical tubes, as a function of the rescaled energy gain $\mu R/\kappa$ associated with helical tubes with rescaled membrane tension $\sigma R^2/\kappa$. The solid purple line is the phase boundary. **b** Rescaled critical radius $r_c/R$ of the tube at the transition from straight to helical as a function of the rescaled surface tension $\sigma R^2/\kappa$. **c** Schematic of the more detailed filament elasticity model, which clusters together the filaments bound in the equatorial (blue) and polar modes (red). Source data are provided as a Source Data file.

for the filaments' bending persistence length, and establish that the membrane-binding energy difference $\mu$ per monomer must be greater than 5 $k_BT$. This is slightly larger than the lower bound on binding energy difference inferred with the first mathematical model, implying that the tubes observed in our experiments are well within the helical tube region of the stability diagram (Fig. 4a). By adding the further assumption that Snf7/Vps24/Vps2 filaments have a bending rigidity close to that of Snf7 homopolymers, i.e., by setting $\ell_p = \ell_p^{Snf7} = 250$ nm[5], we were moreover able to infer their torsional persistence length $\ell_t = 45$ nm, comparable to that of DNA at low tension[21], as well as a binding energy difference $\mu$ of $15 k_BT$ per monomer, suggesting that Vps24 and Vps2 could play an even more important role in the binding of ESCRT-III filaments to lipid membranes.

## Discussion

Our findings support the hypothesis that the assembly of multiple strands of ESCRT-III triggers a buckling transition by increasing the filament's torsion angle and/or torsional rigidity, in addition to bending rigidity[5,10]. A previous theoretical model predicts that flat ESCRT-III spirals without torsional rigidity can tubulate membranes by growing out of plane, provided their bending rigidity is high enough[10]. In the presence of torsional rigidity, the filament in the flat spiral would be pre-constrained (no torsion), and the increase of its torsion angle and/or torsional rigidity when it pairs with additional strands would allow the new composite filament to adopt a conformation closer to its preferred torsion (helical). Hence, under these circumstances, a buckling transition is possible with a lower number of ESCRT-III subunits and with compositional heterogeneity, explaining why our previous model[5] required more subunits than are found at sites of intraluminal vesicle formation[9].

However, we have described a membrane deformation that appears to be a buckling opposite to the direction expected in physiological contexts, such as multi-vesicular body formation[1]. We note that the same filaments could also stabilize the inverse direction as well, yet we cannot observe this on large liposomes because their surface-to-volume ratio will always favor outward deformation.

Considering the helical path of ESCRT-III assemblies, structural studies have identified several membrane-interacting surfaces on the inside[6] and the outside[18] of the helix. We identify here a third surface perpendicular to those, which is required for the mechanical stability of the helical membrane tubes observed here. This may reveal a more complex picture of the filament shape transition involved in membrane deformation. If ESCRT-III subunits change their membrane-binding interface during membrane deformation, this could allow a filament to roll on the membrane and generate torque along the filament axis as another source of membrane strain. This provides a microscopic argument in support of recent coarse-grained simulations, which suggest that torque generation from a polymer rolling on the membrane can lead to both neck formation and scission[22].

Shape buckling and torque may originate from subunits being exchanged for different subunits that bind the membrane with a different preferred orientation. We have previously shown that both subunit turnover and incorporation of different subunits are necessary for ESCRT-III-mediated membrane remodeling[12,23]. In addition, or alternatively, the formation of a secondary membrane-binding filament parallel to the leading strand[12] could change the membrane-binding interface orientation, forcing the membrane to adopt a tubular shape.

Our data did not allow us to establish whether polar and equatorial binding modes reflect different heteropolymer stoichiometries or different conformations of the same proteins forming the heteropolymer, or both. We favor the notion that both filament types contain all three subunits (Snf7, Vps24, and Vps2) as they do not form in the absence of any one of them and different ESCRT-III polymers and copolymers display considerable flexibility[2–5,12,24].

In this study, we show that the different architectures and mechanical properties of ESCRT-III copolymers allow them to stabilize complex membrane shapes. This versatility could well explain the ubiquitous requirement of ESCRT-III as modular membrane-remodeling complex.

## Methods

**Protein expression and purification.** Proteins were expressed from plasmids encoding budding yeast Snf7 (Addgene no. 21492), Vps2 (Addgene no. 21494) and Vps24 (gift from James Hurley), and were purified as previously described[12].

**Liposome preparation.** 1,2-dioleoyl-sn-glycero-3-phosphocholine (DOPC) and 1,2-dioleoyl-sn-glycero-3-phospho-L-serine (sodium salt) (DOPS) were purchased in solution from Avanti Polar Lipids and mixed at the desired molar ratio in chloroform. The lipid mix was dried first under a nitrogen stream and then under vacuum at 30 °C for 1 h before hydration with 100 mM NaCl 20 mM Hepes pH = 7.5. We made large unilamellar vesicles (LUVs) by extrusion of the hydrated lipid films using a Mini Extruder (Avanti Polar Lipids) and polycarbonate filters of pore size 0.2 µm (Whatman).

**Formation of helical membrane tubes.** At 4 °C, in 100 mM NaCl, 20 mM Hepes pH = 7.5, extruded LUVs made from DOPC/DOPS (60/40 mol/mol) (10 mM final) were incubated with 10 µM Snf7 for 1 h, then Vps2 and Vps24 (5 µM each) were added and incubated overnight. For cryo-EM, 4 µL of the sample were deposited on glow-discharged Quantifoil R2/2 200 mesh copper grids and plunge frozen in liquid ethane after a two-sided blot using a FEI Vitrobot. For cryo-ET, we added 10 nm BSA-nanogold (Aurion) to the reaction prior to vitrification. For negative stain EM, the sample was diluted 1/10 in 100 mM NaCl, 20 mM Hepes pH = 7.5 before staining for 30 s with 2% uranyl acetate.

**Formation of protein polymers on bicelles.** We prepared micelles by solubilizing a dried lipid film made from DOPC/DOPS (60/40 mol/mol) at 25 °C in 100 mM NaCl, 20 mM Hepes pH = 7.5, 20 mM CHAPS (3-[(3-Cholamidopropyl)-dimethyl-ammonio]-1-propanesulfonate hydrate, Sigma-Aldrich) at a total lipid concentration of 12 mM. The following protocol is adapted from[25]. In brief, micelles were homogenized by bath sonication and stirring at 25 °C for 1 h before addition of 4 µM Snf7, 2 µM Vps24 and 2 µM Vps2, making sure that the detergent concentration was above its critical micellar concentration after addition of all proteins. The sample was then gradually diluted four-fold over 30 min under agitation at 25 °C and further incubated for 5 h. For cryo-EM, 4 µL of the sample were deposited on glow-discharged Quantifoil R1.2/1.3 300 mesh copper grids and plunge frozen in liquid ethane after a two-sided blot using a FEI Vitrobot.

**Low-resolution EM data collection.** Transmission electron micrographs of negatively stained liposomes and helical tubes were acquired on a FEI Tecnai G2 Sphera LaB₆ at 200 kV using a 4k × 4k FEI Eagle Camera. Vitrified liposomes and helical tubes were imaged in low-dose mode on the same instrument.

**High-resolution cryogenic EM data collection.** 962 Transmission electron cryo-micrographs of helical filaments on bicelles were collected on a FEI Titan Krios XFEG microscope at the University of California San Francisco, USA, equipped with a GIF K2 Quantum System (Gatan) and operated by Serial-EM software at 300 kV. Micrographs were collected at a nominal magnification of ×105,000 in super-resolution mode, corresponding to a super-resolution pixel size of 0.69 Å. Images were collected as dose-fractionated stacks for a total of 80 frames (0.2 s/frame) and total dose of 67.2 electrons/Å². Coma-free beam alignments were performed prior to data collection and automated data collection was conducted with SerialEM[26].

**Cryogenic EM data processing.** Each dose-fractionated image stack was processed using MotionCorr2 and binned by two to yield motion-weighted and dose-weighted images with a pixel size of 1.38 Å/pixel[27]. For the double-filament structures, helical filaments were picked manually using RELION 3.0, segmented into ~90% overlapping segments and additionally binned by two during extraction with a final pixel size of 2.76 Å/pixel (bin4). After rejecting unalignable segments during 2D classification, 35,087 single particle images were processed by 3D classification and 3D auto-refine with helical priors, but without imposing helical symmetry[28,29]. For the single-filament structure, a soft mask was employed to remove one filament from the model and all of the images were re-processed by 3D classification and 3D auto-refine with helical priors and helical symmetry,

using RELION 3.0 software (twist rotation angle (degrees) = 6.7, rise along axis = 10.6 Å). Independent half maps were post-processed using automated procedures. Reported resolutions based on FSC 0.143 criteria[30].

**Electron cryo-tomography data collection**. 73 tilt series were collected on a FEI Titan Krios XFEG microscope at the European Molecular Biology Laboratory, Heidelberg, Germany, equipped with a GIF K2 Quantum System (Gatan) and operated by Serial-EM software at 300 kV. The tilt series were collected using a dose-symmetric scheme[31] ranging ±61° with 2° increments and defoci between −2.5 and −3.5 μm. The nominal magnification was ×65,000 with a calibrated pixel size of 2.14 Å. Images were recorded in counting mode with five frames per tilt angle and a total dose of 120 e$^-$ Å$^{-2}$ (2 e$^-$ Å$^{-2}$ s$^{-1}$ per tilt angle).

**Tomogram reconstruction and subtomogram averaging**. Tilt series of combined frames were aligned using the gold fiducials markers in IMOD[32]. Tomograms were then reconstructed from these aligned tilt series using weighted back-projection in IMOD. Tomograms were binned four times and a 3D Gaussian filter of radius 2 was applied to increase contrast. Tomograms were then filtered using Hide Dust in UCSF Chimera[33], or manually segmented and analyzed using 3dmod from the IMOD suite[32]. We used the Dynamo software for particle extraction and subtomogram averaging[34]. We selected 8150 particle positions along the center of a helical axis of membrane tubes in 17 bin2 tomograms. The table containing these positions was used to extract 160$^3$ pixel particles from bin2 tomograms (bin2 particles). Bin2 particles were used to compute a first reference-free subtomogram average using a single, manually selected, blurred particle as alignment template, yielding a map at 31.9 Å resolution. This first subtomogram average was then used as a template for the alignment of 2037 bin2 particles with a soft elliptical alignment and classification mask focusing on the equatorial filament cluster, yielding a map with a resolution 32.4 Å. Reported resolutions based on FSC 0.143 criterion[30].

**Reporting summary**. Further information on research design is available in the Nature Research Reporting Summary linked to this article.

## Data availability
Data supporting the findings of this manuscript are available from the corresponding authors upon reasonable request. A reporting summary for this Article is available as a Supplementary Information file. The source data underlying Figs. 1h, 2d–g, 3a, b and 4a, b, Supplementary Figs. 2c, 3h and Supplementary Table 1 are provided as a Source Data file. Structural data is available from the Electron Microscopy Data Bank, accession numbers for electron density maps are EMD-10136, EMD-10137, EMD-10138, and EMD-10139.

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

## Acknowledgements
The authors would like to thank Alexander Myasnikov, Arthur Melo and Wim Hagen for help with electron microscopy data collection and processing. The tomography data collection was funded through iNEXT EM HEDC (PID: 6073). J.M.F. acknowledges funding through an EMBO Long-Term Fellowship (ALTF 1065-2015), the European Commission FP7 (Marie Curie Actions, LTFCOFUND2013, GA-2013-609409) and a Transitional Postdoc fellowship (2015/345) from the Swiss SystemsX.ch initiative, evaluated by the Swiss National Science Foundation. A.R. acknowledges funding from the Swiss National Fund for Research Grants N°31003A_130520, N°31003A_149975 and

N°31003A_173087, and the European Research Council Consolidator Grant N° 311536. AR thanks the NCCR Chemical Biology for constant support during this project. L.B. is supported by the "IDI 2016" project funded by the IDEX Paris-Saclay, ANR-11-IDEX-0003-02. M.L. acknowledges support by ANR grant ANR-15-CE13-0004-03 and ERC Starting Grant 677532. M.L.'s group belongs to the CNRS consortium CellTiss. The UCSF Center for Advanced CryoEM is supported by NIH grants S10OD020054 and 1S10OD021741 and the Howard Hughes Medical Institute (HHMI). I.J. was funded by a graduate research fellowship from the National Science Foundation (1000232072) and a Mortiz-Heyman Discovery Fellowship. A.F. is supported by an HHMI Faculty Scholar grant, the American Asthma Foundation, the Chan Zuckerberg Biohub, NIH/NIAID grant P50 AI150464-13 and NIH/NIGMS grant 1R01GM127673-01.

## Author contributions

Conception and design: J.M.v.F. and A.R.; data acquisition, analysis and interpretation: J. M.v.F., L.B., N.T., I.E.J., A.F., M.L., and A.R.; theoretical model: L.B. and M.L.; writing (original draft): J.M.v.F., L.B., M.L., and A.R.; writing (review and editing): J.M.v.F., L.B., N.T., I.E.J., A.F., M.L., and A.R.

## Competing interests

The authors declare no competing interests.
