## [Peer Review File · Nature Communications]

Reviewers' Comments:

Reviewer #1:

Remarks to the Author:

I have carefully read the revised version of the manuscript by von Filscek et al. and I found it has further improved from its original version. In particular i) the authors better explained why they expect the two filaments to be of very similar, if not the same, compositions ii) the geometrical parameters of the filaments have been included iii) the mathematical modelling part in the main text has been rewritten and its connection to the experiments is clearer iv) the connection between the central findings of the paper and possible mechanistic scenarios behind the ESCRT-III function is clearer.

In the meantime, a manuscript with some similar findings has appeared on bioRxiv (Bertin et al. doi.org/10.1101/847319) and it would be helpful to the community if the connection between the two papers has been made, at least by citing each other. Hence I would recommend including the citation to that paper before publishing.

Otherwise I have no further comments, and am happy to support the publication in Nat Comm.

Reviewer #2:

Remarks to the Author:

In the manuscript submitted by Roux and colleagues the structural organization of the ESCRT-III components Snf7 VPS2 and VPS24 is studied in vitro in the presence of membranes, using Cryo-ET. The experimental structural data are then supported by mathematical modelling to provide an explanation to how the ESCRT-III heteropolymer remodels membranes. Although the physiological relevance of these results is yet to be determined, this work is definitely a step forward in understanding the properties of ESCRT-III filaments. I find the revised version of the manuscript suitable for publication in Nature Communications.

REVIEWERS' COMMENTS:

Reviewer #1 (Remarks to the Author):

I have carefully read the revised version of the manuscript by von Filscek et al. and I found it has further improved from its original version. In particular i) the authors better explained why they expect the two filaments to be of very similar, if not the same, compositions ii) the geometrical parameters of the filaments have been included iii) the mathematical modelling part in the main text has been rewritten and its connection to the experiments is clearer iv) the connection between the central findings of the paper and possible mechanistic scenarios behind the ESCRT-III function is clearer.

In the meantime, a manuscript with some similar findings has appeared on bioRxiv (Bertin et al. doi.org/10.1101/847319) and it would be helpful to the community if the connection between the two papers has been made, at least by citing each other. Hence I would recommend including the citation to that paper before publishing.

Otherwise I have no further comments, and am happy to support the publication in Nat Comm.

Point-by-point response to reviewer #1:

- We have added a citation of the preprint from Bertin et al.

Reviewer #2 (Remarks to the Author):

In the manuscript submitted by Roux and colleagues the structural organization of the ESCRT-III components Snf7 VPS2 and VPS24 is studied in vitro in the presence of membranes, using Cryo-ET. The experimental structural data are then supported by mathematical modelling to provide an explanation to how the ESCRT-III heteropolymer remodels membranes. Although the physiological relevance of these results is yet to be determined, this work is definitely a step forward in understanding the properties of ESCRT-III filaments. I find the revised version of the manuscript suitable for publication in Nature Communications.

Point-by-point response to reviewer #2:

- n/a